# 16S rRNA of Mucosal Colon Microbiome and CCL2 Circulating Levels Are Potential Biomarkers in Colorectal Cancer

**DOI:** 10.3390/ijms221910747

**Published:** 2021-10-04

**Authors:** Carmela Nardelli, Ilaria Granata, Marcella Nunziato, Mario Setaro, Fortunata Carbone, Claudio Zulli, Vincenzo Pilone, Ettore Domenico Capoluongo, Giovanni Domenico De Palma, Francesco Corcione, Giuseppe Matarese, Francesco Salvatore, Lucia Sacchetti

**Affiliations:** 1Department of Molecular Medicine and Medical Biotechnologies, University of Naples Federico II, 80131 Naples, Italy; carmela.nardelli@unina.it (C.N.); nunziato@ceinge.unina.it (M.N.); capoluongo@ceinge.unina.it (E.D.C.); giuseppe.matarese@unina.it (G.M.); salvator@unina.it (F.S.); 2CEINGE Biotecnologie Avanzate S.C.a R.L., 80131 Naples, Italy; setaro@ceinge.unina.it; 3Task Force on Microbiome Studies, University of Naples Federico II, 80131 Naples, Italy; 4Institute for High Performance Computing and Networking (ICAR), National Research Council (CNR), 80131 Naples, Italy; ilaria.granata@icar.cnr.it; 5Laboratory of Immunology, Institute of Endocrinology and Experimental Oncology, Consiglio Nazionale Delle Ricerche (IEOS-CNR), 80131 Naples, Italy; f.carbone@ieos.cnr.it; 6Neuroimmunology Unit, IRCCS Fondazione Santa Lucia, 00143 Roma, Italy; 7Digestive Endoscopy Unit, Gaetano Fucito Hospital, Mercato San Severino, 84085 Salerno, Italy; claudio.zulli@sangiovannieruggi.it; 8Department of Medicine and Surgery, AOU San Giovanni di Dio e Ruggi D’Aragona, University of Salerno, 84084 Salerno, Italy; vpilone@unisa.it; 9Department Clinical Medicine and Surgery, Federico II University of Naples, Via Pansini 5, 80131 Naples, Italy; giovanni.depalma@unina.it; 10Department of Public Health, School of Medicine and Surgery, University of Naples Federico II, Via Pansini 5, 80131 Naples, Italy; francesco.corcione@unina.it

**Keywords:** CCL2 [chemokine (C-C motif) ligand 2], mucosal colon microbiome, 16S rRNA gene, obesity, colorectal cancer

## Abstract

Colorectal cancer (CRC) is one of the most common malignancies in the Western world and intestinal dysbiosis might contribute to its pathogenesis. The mucosal colon microbiome and C-C motif chemokine 2 (CCL2) were investigated in 20 healthy controls (HC) and 20 CRC patients using 16S rRNA sequencing and immunoluminescent assay, respectively. A total of 10 HC subjects were classified as overweight/obese (OW/OB_HC) and 10 subjects were normal weight (NW_HC); 15 CRC patients were classified as OW/OB_CRC and 5 patients were NW_CRC. Results: *Fusobacterium nucleatum* and *Escherichia coli* were more abundant in OW/OB_HC than in NW_HC microbiomes. Globally, *Streptococcus intermedius*, *Gemella haemolysans*, *Fusobacterium nucleatum*, *Bacteroides fragilis* and *Escherichia coli* were significantly increased in CRC patient tumor/lesioned tissue (CRC_LT) and CRC patient unlesioned tissue (CRC_ULT) microbiomes compared to HC microbiomes. CCL2 circulating levels were associated with tumor presence and with the abundance of *Fusobacterium nucleatum*, *Bacteroides fragilis* and *Gemella haemolysans*. Our data suggest that mucosal colon dysbiosis might contribute to CRC pathogenesis by inducing inflammation. Notably, *Fusobacterium nucleatum*, which was more abundant in the OW/OB_HC than in the NW_HC microbiomes, might represent a putative link between obesity and increased CRC risk.

## 1. Introduction

Colorectal cancer (CRC) is one of the most common malignancies in the Western world [1]. Accordingly, in Italy, CRC is the second/third most frequent neoplasia in males and in females, respectively, and among the most common causes of cancer death [2]. Up to now, various genomic subtypes of CRC have been identified that vary by side (ascending/right, descending/left, etc.) intestinal tract, and show different histology and outcomes [3]. Several risk factors for CRC have been, until now, described including diet, smoking, alcohol, aging, obesity and, recently, intestinal microbiota has also been implicated in its insurgence and/or in the response to therapy [4,5,6,7]. The gut is populated by trillions of microbial cells (mainly bacteria but also archaea, fungi, protozoa and viruses) that, by symbiotic interaction with their host, play important roles in human health, contributing to nutrient absorption, immune response, metabolism and to maintaining the integrity of the intestinal barrier [8]. Alterations in the gut microbiota and in their metabolic products might induce dysfunctions in the gut epithelial barrier, thereby leading to aberrant immune activation and to local and systemic inflammation [9]. It is worth remembering that inflammation is one of the initiating processes of carcinogenesis, and that the chemokines that guide the immune cells to the tumor site have been associated with increased tumor growth and progression [10]. Among cytokines, chemokine (C-C motif) ligand 2 (CCL2), also named monocyte chemoattractant protein 1 (MCP1), is produced by tumor cells and by various other cells in the host microenvironment, including adipocytes, and was reported upregulated in various tumors, including CRC, and in obesity [11,12]. This chemokine has been previously involved in tumor promotion under obese condition [13] and elevated circulating CCL2 levels have been associated with bone metastasis in prostate cancer patients [14]. Consequently, we specifically chose to measure the levels of CCL2 to investigate its putative role as a diagnostic biomarker linking inflammation/obesity to CRC. As mentioned before, gut dysbiosis has been described in several conditions such as metabolic, autoimmune, inflammatory bowel diseases and obesity, and in different tumors, including CRC [6,15,16]. Microbes play important roles in obesity. Previously, our group investigated the duodenal metatranscriptome in human obese and lean subjects, highlighting an obese-associated gut dysbiosis and human and bacterial genes both contributing to relevant dysfunctions in important metabolic pathways [17,18]. Regarding CRC alterations until now reported in the gut microbiota community or in single bacterial species in association with the tumor [19], in presence/absence of various risk factors among which increased adiposity, they have been mostly studied in murine models and/or in human fecal samples, but the human mucosal colon microbiome has still scarcely been investigated [20,21,22,23]. In this study we: (a) typed the mucosal colon microbiome in CRC patient tumor lesioned tissue (CRC_LT) and CRC unlesioned tissue (CRC_ULT) from surgical resection in lean and overweight/obese CRC patients by 16S rRNA sequencing; (b) compared the CRC-associated microbiomes with those found in mucosal biopsies (from colonoscopy) of healthy lean and overweight/obese subjects; and (c) measured the CCL2 blood levels, as a biomarker of immune inflammation in both healthy and CRC subjects.

Our aims were: (a) to investigate the differences in the mucosal colon microbiome between healthy and CRC subjects, with and without the additional contribution of the overweight/obesity co-morbidity; (b) to explore if tumor-associated microorganisms are also present in the nearby non-cancerous region; and (c) to look for any association between significantly CRC-associated bacteria and the systemic inflammation, as assessed by CCL2 circulating levels. Our data in providing further insight into the CRC-associated microbiome alterations might hopefully contribute to future potential microbiome- and/or cytokine-based diagnostic tools in the CRC management.

## 2. Results

### 2.1. Mucosal Colon 16S rRNA Analysis

In the present study we analyzed the microbial composition of the mucosal colon from healthy individuals (HC) and from CRC patients. First, we analyzed the microbial composition in HC. In Appendix A the alpha diversity measured by Shannon diversity and Chao1 richness indices is shown. No statistically significant difference in species richness (number of species) was detected in normal weight HC (NW_HC) microbiomes compared to overweight/obese HC (OW/OB_HC) microbiomes: NW is defined as BMI < 25 kg/m^2^ and OW/OB is defined as BMI ≥ 25 kg/m^2^. No statistically significant difference in species richness was detected in the microbiomes of females compared to males.

The composition of the difference microbial groups (beta-diversity analysis) was evaluated by PERMANOVA (Adonis), ANOSIM and PERMDISP2 (beta-dispersion) tests. Representation of sample distances through Nonmetric Multidimensional Scaling (NMDS) and the results of the associated statistical tests are shown in Figure 1A,B. The Adonis and ANOSIM tests indicate that the microbiomes of NW_HC and OW/OB_HC were significantly different (Adonis *p*-value = 0.003; ANOSIM *p*-value = 0.005), whereas no significant difference was found using the beta-dispersion test. No significant difference was found by any of the tests when grouping the samples by gender. Taken together, these analyses indicate that there is a significant difference in the compositions of microbial groups between NW_HC and OW/OB_HC, but not between females and males.

Four dominant phyla characterized the mucosal colon microbiome of our healthy control cohort, Actinobacteria, Firmicutes, Bacteroidetes and Proteobacteria, in decreasing relative abundances (Appendix A). Differential abundance analysis highlighted the different taxa that were significantly different (adjusted *p*-value < 0.05) between the OW/OB_HC and NW_HC samples from phylum to species level (Table 1: note that taxa that were not significantly different between the OW/OB_HC and NW_HC samples are not listed).

In detail, Fusobacteria and Proteobacteria phyla were more abundant, Firmicutes was less abundant phyla in OW/OB_HC than in NW_HC, whereas Bacteroidetes and Actinobacteria showed similar abundances in both groups (Figure 2A and Appendix A).

At species level, *Acinetobacter lwoffii*, *Escherichia coli* and *Haemophilus parainfluenzae* (Proteobacteria), *Fusobacterium nucleatum* (Fusobacteria), resulted more abundant, and *Bacteroides dorei* (Bacteroidetes), *Subdoligranulum variabile*, *Bifidobacterium longum*, *Peptoclostridium difficile* and *Ruminococcus torques* (Firmicutes) less abundant in OW/OB_HC than in NW_HC (Table 1).

Secondly, we analyzed the differences in colon microbiome of CRC patients, in particular comparing the tumor lesioned (CRC_LT) to the unlesioned (CRC_ULT) tissues. In Appendix A the alpha diversity measured by Shannon index is shown, considering (A): CRC_LT and CRC_ULT samples divided by overweight/obesity comorbidity; (B) CRC_LT and CRC_ULT samples grouped by the different tumor location (ascending colon AC, rectum R and sigma-descending colon S_DC); (C) the samples grouped for tumor stages (AJCC staging) and divided by the location. Only in the sigma-descending colon tract statistically significant differences in microbial richness and diversity were detected comparing I_II vs. III_IV tumor stages (*** *p* < 0.005). Furthermore, the PCoA of weighted UniFrac distance showed that the distributions of microbiomes in CRC patients were significantly separated on the basis of tumor location and stage (PERMANOVA (Adonis) test *p* < 0.05) (Figure 3). In fact, the test highlights that the centroids of each group are significantly different from each other (*p* < 0.05).

Comparison of bacterial species differently present in the several colon tracts are detailed in Appendix A.

*Akkermansia muciniphila*, *Granulicatella adiacens*, *Streptococcus intermedius* and *Gemella haemolysans* were significantly more abundant, whereas *Alistipes* spp, *Bacteroides* spp. and *Parabacteroides distasonis* were significantly less abundant in the sigma-descending colon than in ascending colon. *Fusobacterium nucleatum* and *Bacteroides fragilis* were both more present in sigma-descending colon and ascending colon than in rectum. Regarding the comparison of microbiomes in CRC_LT versus CRC_ULT samples, *Fusobacterium nucleatum* and *Bacteroides fragilis* were significantly more abundant in CRC_NW_LT than in CRC_NW_ULT (8.82 and 5.58 Log2FC, respectively) (Figure 4). Further, slight increased abundances of these two bacteria, even if not at significant level, were observed comparing CRC_OW/OB_ LT with CRC_OW/OB_ ULT samples (Figure 4).

Finally, we compared the microbiomes of CRC patients with HC subjects. The PCoA of weighted UniFrac distance showed a significant separation between the two groups (PERMANOVA (Adonis) test, *p* = 0.001) (Figure 5A,B). The significant beta-dispersion (*p* = 0.022) indicated that two groups were not homogenously dispersed. Separation resulted significant at ANOSIM test (*p* = 0.001) when we compared CRC_ULT/CRC_LT vs. HC.

The differentially abundant species between CRC and HC, resulting from multiple comparisons, are reported in Table 2. The most abundant species in CRC vs. HC as well as in CRC_LT vs. HC were: *Streptococcus intermedius* (Firmicutes)*, Gemella haemolysans* (Firmicutes), *Fusobacterium nucleatum* (Fusobacteria), *Bacteroides fragilis* (Bacteroidetes) and *Escherichia coli* (Proteobacteria). Interestingly, these bacteria were also more abundant in CRC_ULT vs. HC, except the *Bacteroides fragilis*, suggesting a microbial profile significantly associated with CRC both in tumor lesioned and unlesioned tissues. *Ruminococcus torques*, previously found significantly less abundant in OW/OB_HC than in NW_HC, was also less abundant in CRC vs. HC and in CRC_ULT vs. HC (Table 2). In Appendix A, the boxplots showing the comparisons between CRC and HC for the several significant bacterial species are reported.

### 2.2. Impact of Aging on Microbiome Composition

To exclude that the CRC associated microbiome profile could be influenced by the different age range of our HC and CRC subjects, we used the median age (64 years) of our entire cohort of 40 samples as cutoff to categorize the individuals as adult (≤64 years) or old (>64 years). The alpha diversity measured by Shannon index (Appendix A) showed no significant difference in the abundance and evenness of the species present within the microbial community of the two differently aged groups, whereas the number of taxa (species richness) measured by Chao index (Appendix A) was not statistically significant within HC and within CRC ≤ or >64 years, but significantly higher in CRC compared to HC in both groups of different ages. The beta-diversity analysis is shown in Appendix A and highlighted that the centroids of each group were significantly different from each other (PERMANOVA (Adonis) *p*-adjusted = 0.003). In Appendix A, the box plots of microbial profiles in the two groups of HC and CRC subjects (Log normalized counts) obtained at phylum and species levels (panel A and B, respectively) are reported. The counts were modeled considering the age categorization as a covariate before performing the differential abundance test using the DeSeq2 package. The latter test highlighted that Actinobacteria and Fusobacteria phyla (adjusted *p*-value < 0.001) and the species *Streptococcus intermedius*, *Gemella haemolysans*, *Fusobacterium nucleatum*, *Escherichia coli* (adjusted *p*-value < 0.001) were significantly different between CRC and HC, independently from the age differences between the two groups.

### 2.3. Association between CCL2 Blood Levels in CRC Patients and Healthy Controls and the Abundances of Bacteria Significantly CRC-Associated

We found significantly higher mean blood levels of CCL2 in CRC than in HC samples (102 vs. 38 pg/mL, *p* = 0.009). In order to verify the association between CCL2 blood levels and the abundance of bacteria found significantly associated with CRC, we performed linear multiple regression analysis building a model for each of the selected bacteria including the category variable of the tumor presence. The multiple tests allow to consider in the same model the association of the two variables (bacterium abundance and presence of tumor) to the CCL2 levels. First, we checked that regardless of the age of the study groups, CCL2 plasma levels were significantly associated with tumor presence (*p* = 0.00343) (Figure 6A–C). Among the bacteria resulting significantly associated with the tumor (more than two folds increased abundance with respect to HC) in our study, *Fusobacterium nucleatum*, *Bacteroides fragilis* and *Gemella haemolysans* were found to be significantly associated with CCL2 blood levels (*p* = 0.030, 0.050 and 0.001, respectively) (Figure 6A–C), while *Streptococcus intermedius* was not associated. When the *p*-values are significant, we can reject the null hypothesis and conclude that *F. Nucleatum* or *B. fragilis* abundance, together with the presence of tumor, likely influence rates of CCL2. In the case of the model built with the *G. Haemolysan* abundance and tumor presence variables, it tells that the estimated increase in CCL2 associated to *G. haemolysans* is higher than to the other bacteria and, contrary to *F. Nucleatum* and *B. fragilis*, the increase in CCL2 levels is totally influenced and explained by the increase in bacterium abundance, as their association is stronger, making the presence of tumor irrelevant.

## 3. Discussion

In the last years, it has been suggested that gut dysbiosis could be involved in the onset of obesity but also of other metabolic diseases and gastrointestinal cancers, including CRC [15]. The hypothesized biologic mechanism linking excess body fat to cancer risk is that adipocytes secrete an array of bioactive signaling molecules including pro-inflammatory adipokines and cytokines that, by inducing chronic low-grade inflammation, may stimulate cancer development [24]. To support the association between obesity and colorectal cancer risk, the observational epidemiologic evidence indicates the 10–30% magnitude of relative risk increases for BMI ≥ 25 kg/m^2^ versus BMI < 25 kg/m^2^ [25].

Here, we first investigated the mucosal colon microbiome in two groups of healthy subjects and CRC patients divided by weight in NW and OW/OB according to their BMI< or ≥25 kg/m^2^, respectively, to highlight if there were obese-associated bacterial alterations, possibly in common between HC and CRC, that might likely address the increased risk of cancer development in obesity. Second, we explored if CRC-associated bacterial abundances differed between cancerous and non-cancerous tissue in the same patient. Finally, we evaluated if the bacterial alterations found in CRC patients correlated with systemic inflammation, as evaluated by the circulating levels of the CCL2 cytokine marker previously found increased in several cancers and in obesity [11,12].

### 3.1. Mucosal Colon Microbiome Was Different between NW and OW/OB Control Subjects

In the mucosal microbiome of our HC subjects, we found significantly reduced abundance of *Bacteroides dorei* (Bacteroidetes), *Bifidobacterium longum* (Actinobacteria), *Peptoclostridium difficile*, *Subligranulum variabile* and *Ruminococcus torques* (Firmicutes) species in OW/OB_HC with respect to NW_HC. Interestingly, we also found increased abundances in OW/OB_HC with respect to NW_HC of *Fusobacterium nucleatum* (Fusobacteria) and *Escherichia coli* (Proteobacteria); both species, often described in CRC-microbiomes and known to produce metabolites, able to drive inflammation and alterations of the intestinal barrier function [19,26]. *Fusobacterium nucleatum* has also been previously reported to be more abundant in obese than in lean Japanese subjects [27]. Globally, contrasting data are reported in literature regarding the obesity-associated dysbiosis, independent of the different studies and the several factors influencing the gut microbiome (i.e., animal/human study models, geographical area of the studied cohorts, dietary habits or treatments, studied samples (i.e., faeces, colonic mucosal biopsies or other intestinal tracts) and analytical methods used to study microbiome) [23,28]. Usually, but not always, increased relative abundance of Firmicutes and reduced relative abundance of Bacteroidetes were found in the obese-associated microbiomes [29,30,31,32]. However, as a consequence of the accumulation of data, the ratio Firmicutes/Bacteroidetes relative abundance was no longer confirmed to be a clear biomarker of obesity susceptibility [33,34]. Moreover, a recent metanalysis using five publicly available stool and tissue-based 16S rRNA and whole genome sequencing data set, identified *Bacteroides* spp. and *Bifidobacterium* spp. as the species most often able to differentiate individuals with and without obesity [35]. In accordance with the latter report, we found significantly reduced abundances of *Bacteroides* spp. and *Bifidobacterium* spp. in our OW/OB_HC with respect to NW_HC. Interestingly, depletion of short chain fatty acids (SCFA)-producing bacteria such as *Bifidobacterium longum* has been reported to promote colorectal carcinogenesis in obese patients [36].

### 3.2. Differences in Mucosal Colon Microbiome between Tumor Lesioned and Unlesioned Tissues of CRC Patients

In our CRC patients divided by weight, we detected significant increased relative abundances of *Fusobacterium nucleatum* and *Bacteroides fragilis* in colon microbiome from LT_NW with respect to ULT_NW and no significantly different taxa between LT_OW/OB and ULT_OW/OB. However, in CRC OW/OB tissues, *Fusobacterium nucleatum* and *Bacteroides fragilis* showed comparable abundances to those found in LT_NW. The latter finding suggests two observations: First, that contrary to what happens in CRC-NW patients, in the presence of overweight/obesity as co-morbidity, the CRC-associated bacterial alterations are not restricted to the lesioned area, in agreement with recent literature data [21]; second, that the increased *Fusobacterium nucleatum* abundance detected in OW/OB_HC with respect to NW_HC subjects, might play a weak effect on mediating CRC risk. Consistently, recent data reported that in individuals with obesity, *Fusobacterium nucleatum* showed strong prediction (log2 OR) of CRC [35]. Moreover, *Bacteroides fragilis* was the only species in intestinal microbiomes reported as being consistently enriched in a metanalysis of metagenomes of CRC patients worldwide [37]. Anyway, we are aware that the number of our samples is limited so our results have to be taken as an indication and need further experimental validation.

PCoA revealed differences in CRC mucosal colon microbiome depending on the intestinal tumor location and on the tumor stage. In particular, *Fusobacterium nucleatum* and *Bacteroides fragilis* were less abundant and *Bifibacterium longum* was more abundant in rectal than in sigma-descending colon and ascending colon tumors, but the abundances of these two bacteria did not differ between sigma-descending colon and ascending colon tumor locations. *Akkermansia muciniphila* was highly significantly abundant in CRC patients with rectal and sigma-descending tumors, whereas *Gemella haemolysans* and *Streptococcus intermedius* abounded, and *Alistipes* spp. were reduced in sigma-descending colon with respect to rectum and ascending colon. In this regard, we would underline that although *Akkermansia muciniphila* is considered a protective microbe, high abundance of this bacterium was previously described in individuals with rectal and distal colon cancer [38]. Nonetheless, the role of this species needs to be further investigated in terms of any possible mechanistic effect on the cancer. In agreement to our data, significant differences in microbial alterations between distal and proximal CRC have been previously described [22]. Although PCoA highlighted in our study differences in microbial patterns depending on tumor stage, the differently staged tumors were randomly located in the different colon tracts. Further, it should be remembered that right and left sides of the colon have distinct embryologic origins and biology, different gene expression profiles and methylation patterns, different tumor suppressors and oncogene mutations that are all biologic distinctions able in influencing microbial composition, host-microbiome interaction and impacting tumor prognosis [3]. That said, considering the small CRC cohort here studied and because we lacked genetic tumor characteristics of our patients, we could not discriminate among the significant stage-associated microbial changes to those actually correlated to the stage of the tumor or to other influencing characteristics.

### 3.3. Mucosal Colon CRC Microbiome Differs Significantly from That of Healthy Controls

Mucosal colon microbiomes resulted significantly different between our CRC patients and HC subjects. Further, PCoA analysis highlighted that both CRC_ULT and CRC_LT associated microbiomes were similar to each other and significantly different from the HC microbiomes. In particular, the CRC-associated microbiome was characterized by increased abundances of *Streptococcus intermedius, Gemella haemolysans, Fusobacterium nucleatum, Bacteroides fragilis* and *Escherichia coli*. All these bacteria were also more abundant in CRC_ULT vs. HC, except the *Bacteroides fragilis.* Our findings support that microbial change in CRC colon patients occurs as a continuum from the CRC unlesioned (adjacent normal) tissue to the CRC lesioned tissue. However, these results deserve further validation in a larger than the present cohort.

So far, several mechanisms by which bacteria can promote an environment that favors the development of cancer have been described, among which are the generation of toxic metabolites that may be pro-carcinogenic, DNA damage, induction of cell proliferation, alteration of the intestinal barrier, post-transcriptional modification via altered microRNA expression and inflammatory capability [6,26].

In particular, among CRC-associated bacteria in our cohort, the *Fusobacterium nucleatum* has been frequently found enriched in patients with colorectal cancer [23]. It has been correlated with DNA methylation of genes within the inflamed colonic mucosa which enhances tumorigenesis [39,40]. *F. nucleatum* is also capable of inducing inflammation and alteration in the intestinal barrier by producing lipopolysaccharide, reactive oxygen and nitrogen species [41]. Fusobacteria were also used to expand their ecological niches by entering the cell and inducing proliferation, so promoting colorectal cancer [42,43]. *Bacteroides fragilis* is a pathogenic bacterium able to colonize the intestinal epithelium with pro-carcinogenic capabilities and it produces reactive oxygen species that can damage host DNA so contributing to colon cancer [44]. Further, the enterotoxigenic *Bacteroides fragilis,* which produces the *Bacteroides fragilis* toxin, was found enriched in CRC tissues compared to tissues from controls or adjacent non-cancerous tissue [45]. *B. fragilis* toxin activates Wnt and NF-kβ signaling pathways and increases the release of pro-inflammatory molecules from epithelial intestinal cells [44]. Accordingly, we found higher *B. fragilis* abundance in CRC microbiome than in HC or ULT tissues, although further experiments need to establish whether the *B. fragilis* species found in our CRC cohort is the enterotoxic one.

*Escherichia coli* is able to produce toxin as colibactin with oncogenic potential by causing DNA crosslinks and double strand DNA breaks [46]. Mucosa-associated *E. coli* increased abundance has been reported in CRC tissue and correlates with tumor stage and prognosis [47].

In our CRC associated microbiome, along with *Fusobacterium nucleatum, Bacteroides fragilis* and *Escherichia coli*, we also found as new players *Streptococcus* spp. and *Gemella* spp. These latter bacteria were recently reported in CRC microbiome and hypothesized to produce oxidative stress and inflammation [48,49,50,51].

### 3.4. Circulating Levels of CCL2 in CRC and Healthy Controls

We found significantly higher circulating levels of CCL2 in CRC patients than in control subjects. In agreement to our data, several chemokines, among which was CCL2, have been previously associated with increased tumor growth and progression [11]. Further, inhibition of CCL2 signaling was shown to reduce cancer cell metastasis in tumor bearing mice, so CCL2 might be a potential target for the cancer treatment [52]. More elevated circulating levels of CCL2 were measured in aged mice than in young mice [53] and age-related changes in the microbiome might potentially contribute to the pro-inflammatory environment associated with the aging process [54]. In our study, we found that CCL2 circulating levels correlated significantly with the tumor presence and with the abundance of *Fusobacterium nucleatum, Bacteroides fragilis* and *Gemella haemolysans* in the colon of our CRC patients. Moreover, the CCL2 increase was almost independent from the age of the CRC patients since we found only slightly increased CCL2 levels in control subjects with increasing age with respect to those observed in CRC patients at any age. Previously, increased abundance of *F. nucleatum* has been associated to a high expression of pro-inflammatory cytokines in colonic tissue from CRC patients [7]. On the other hand, *Streptococcus intermedius* does not appear to be a predictive factor of a CCL2 increase. In conclusion, based on our data, we hypothesize that the CRC-associated microbiome directly or by releasing bacteria-derived substances (i.e., LPS, ROS, etc.) might promote intestinal lumen inflammation and induce the CCL2 transcription by cells in the tumor microenvironment. In support of our hypothesis, recent findings link CCL2 to the inflammation and cancer pathogenesis and provide evidence that CCL2 production in tumors is due to complex interactions between tumor and non-tumor cells, both cells contributing to the high tumor-associated CCL2 levels [10,11]. However, we cannot rule out that other potential confounders covariates could contribute in increasing CCL2 levels in CRC and further studies should confirm our results.

In this study there are some limitations. The number of the enrolled CRC and control subjects was relatively small and the mean age of CRC patients, as expected, was higher than that of healthy controls. Nevertheless, after dividing our cohort in ≤ or >64 years, we confirmed that the CRC associated microbiome was independent from age of the subjects. Further, all individuals were from the same geographical area, 50% female, none were hospitalized prior to surgery and all reported following the Mediterranean diet, likely thereby reducing some of the major factors affecting the colon microbiome in aged patients. When cancer patients were divided by tumor location, the number of subjects in each subgroup is low and the differences observed in microbiome profile/location, even if comparable between tumor lesioned and un-lesioned tissue in each patient, should be taken with caution and confirmed in a larger number of patients. Further, the comparison between increases in CCL2 circulating levels and bacterial species abundances in colon observed in CRC patients indicates an association but not a causal link among these variables. For this purpose, additive experiments (i.e., stimulation-like experiments in cellular models with CRC-associated bacterial species) are necessary to demonstrate the inflammatory effect of CRC-associated bacteria. Despite this, ours is one of the first human association studies between the increased CCL2 circulating levels and CRC, previous studies being mostly performed in murine models.

One merit of our study is to have compared the mucosal colon microbiome, not only in the cancerous and non-cancerous tissues of each patient, but also with a control group of the same geographical area and to have also taken into consideration the presence of overweight/obesity as co-morbidity for the onset of cancer.

## 4. Materials and Methods

### 4.1. Patients and Controls

We studied two groups of subjects. Study group 1 (*n* = 20) included healthy controls (HC) (50% female; mean age = 53.2 years) recruited among those undergoing colonoscopy as part of their diagnostic path, in whom the histology revealed a normal mucosa or diverticulitis and the absence of any cancer lesion. Study group 2 (*n* = 20) included CRC patients (50% female; mean age = 69.4 years) that were consecutively recruited among those scheduled for tumor resection.

The study groups were enrolled at the Surgery and Medicine Department of the “Università degli Studi di Salerno” and at the Surgery and Medicine Department of the “Università degli Studi di Napoli Federico II” between 2019 and 2020. Exclusion criteria for both groups were inflammatory bowel disease or irritable bowel syndrome and antibiotics, pro- and pre-biotics, antiviral or corticosteroid treatments in the two months prior the collection of the samples. General characteristics of the CRC group, including tumor location and stage, and of the control group are reported in Appendix A. The clinical and anamnestic data of each subject were collected by an expert clinician. All subjects reported to follow a Mediterranean diet; 20% CRC and 45% HC were smokers. The following diseases/drugs assumption were reported in the CRC patients: hypertension, (25%)/antihypertensive/diuretics; diabetes, (25%)/metformin; cardiomyopathy, (25%)/cardioaspirin; neuropathy, (10%)/edoxaban. All the enrolled subjects gave their informed consent to participate in the study, according to the Helsinki Declaration 2013.

The study was approved by the Ethics Committee of the “Università degli Studi di Napoli Federico II” authorization no. 318/20, 9 September 2020, and the “Università degli Studi di Salerno”, authorization n. 50, 15 July 2015; amendment n. 141070, 26 November 2019.

### 4.2. Sample Collection and Storage

We collected from each enrolled individual a blood sample for biochemical investigations and one or two colon biopsies from HC or CRC patients, respectively. The colon biopsies were taken during surgery (CRC-ULT and CRC-LT) for study group 1, or during colonoscopy (HC) for study group 2. Colonoscopy was performed under sterile conditions to avoid contamination according to quality indicators suggested from Garborg K et al., 2017 [55]. Biopsies were immediately cooled in dry ice and stored at −80 °C within 15 min until DNA isolation for the following microbiome analysis. The same standard procedure was performed in the two clinical centers.

### 4.3. CCL2 Assay

Circulating CCL2 levels in HC and CRC patients were measured in a fixed volume of sample (25 µL) by using Human ProcartaPlex Mix & Match 22-plex (Thermo Fisher Scientific, Waltham, MA, USA) kit, according to manufacturer’s instructions. The assay was run on a Luminex 200 system (Luminex Corporation, Austin, TX, USA). Data were fit to a nine-point standard curve generated using the five-parameter logistic regression model with the xPonent 3.1 software. Extrapolated values were used for samples that exhibited fluorescent signals above background (i.e., above blank samples) but below the standard curve range. CCL2 levels were represented as mean values and *t*-test was performed to compare the CCL2 levels between HC and CRC patients, a *p*-value < 0.05 was considered statistically significant.

### 4.4. DNA Isolation and 16S rRNA Sequencing

Each biopsy (~5 mg) was pulverized with a standard liquid nitrogen pre-chilled mortar and pestle. This powder was transferred to a 1.5 mL tube with 1 mL of Phosphate Buffered Saline (PBS) (Thermo Fisher, Waltham, MA, USA). The homogenate was vortexed vigorously using Vortex-Genie 2 (Scientific Industries, Bohemia, NY, USA). Subsequently, microbial DNA was isolated by using QIAamp DNA Microbiome Kit (Qiagen, Venlo, The Netherlands), according to manufacturer’s instructions. All extractions were performed in a pre-PCR designated room using a biosafety cabinet. The yield and the quality of extracted microbial DNA were determined using Qubit dsDNA HS (High Sensitivity) Assay Kit (Invitrogen Co., Life Sciences, Carlsbad, CA, USA) and TapeStation (Agilent Technologies, Santa Clara, CA, USA), respectively. The amplification of the hypervariable V3-V4-V6 regions of the bacterial 16S rRNA was performed using the Microbiota solution B (Arrow Diagnostics, Genova, Italy), according to the manufacturer’s instructions. The quality and quantity of libraries were evaluated by TapeStation system and Qubit dsDNA HS Assay, respectively, to prepare the equimolar libraries pool. Sequencing was performed on MiSeq Illumina^®^ sequencing platform (Illumina, CA, USA) using V2 500 cycles reagent. Possible contaminations during the above procedural steps were verified using a negative control and the repeatability of the sequencing procedures assessed using a standard control (Gut Microbiome genomic Mix-ATCC MSA-1006, LGC Standard, Milan, Italy) processed simultaneously to the patients’ samples.

### 4.5. Reads Processing

The raw paired ends FASTQ files of 60 samples, 6 negative controls and 6 standards obtained from 6 sequencing runs (ranging from 14,355 to 150,520 reads) were quality filtered and trimmed (5858–101,620 reads left) by filterAndTrim function of DADA2 R package v. 1.12.1 [56]. Given the length of the V3-V6 amplicon (721 bp) and the reads length (2 × 250), the paired mates did not overlap. In these cases, it is often advised to use only first reads for taxonomy annotation, thus losing information from the second reads. A recent strategy, called JTax, was proposed by Liu et al. [57] based on joining non-overlapping reads and rearranging the reference sequences through primer sites accordingly. JTax was then used to join the filtered paired FASTQ files and to rebuild 16S-Udb, a reference unified database obtained by merging non-ambiguous, fully annotated, full-length 16S rRNA sequences from Greengenes, SILVA and Ribosomal Database Project (RDP) databases [58].

Clustering of 16S reads into operational taxonomic units (OTUs) was performed by using USEARCH v. 11.0.667 [59] as follows: (1) low quality reads of all joined FASTQ were filtered (command: fastq_filter); (2) filtered reads (ranging from 1636 to 46,866) were then de-duplicated into unique reads (command: fastx_uniques); (3) unique reads were clustered into OTUs with a 97% identity (command: cluster_otus, option: -minsize 2); (4) all reads were aligned against the OTU representative sequences (command: usearch_global; options: identity cutoff 0.97, strand both); and (5) taxonomy of query OTU representative sequences was predicted (command: sintax; options: sintax cutoff 0.7, strand both).

The OTUs clustering and assignment steps were repeated for the three analyses, HC, CRC and HC + CRC samples, in order to better adapt the pipeline and the analyses to each specific case study.

### 4.6. Bioinformatics and Statistical Analysis

OTUs table, assigned taxa table and samples metadata were used to build the phyloseq-class experiment-level objects through phyloseq package v. 1.28.0 [60] (HC: 901 taxa and 20 samples; CRC: 2968 taxa and 40 samples; HC + CRC: 3010 taxa and 60 samples).

Alpha-diversity of samples, that measures both the richness and diversity of species within a group, was calculated on taxa that were observed at least once, using alpha function of the Microbiome package v. 1.6.0 [61]. In particular, we used the following indices: CHAO1 that measures the observed richness (number of taxa) and Shannon’s index that accounts for both abundance and evenness of the species present within a microbial community.

Beta-diversity, that measures the differences in microbiome composition between groups, was calculated using the weighted unique fraction metric (UniFrac) that measures the distance on taxa counts table filtered keeping those having counts >1 in at least 10% of samples and then normalized through regularized logarithm transformation (DESeq2 v. 1.24.0 [62]). UniFrac phylogenetic distances are based on the fraction of branch length shared between two communities within a phylogenetic tree constructed from the 16S rRNA gene sequences from all communities being compared. With weighted UniFrac, branch lengths are weighted based on the relative abundances of lineages within communities (community structure).

PERMANOVA (Adonis), ANOSIM and PERMDISP2 (beta-dispersion) tests were performed through Vegan package v. 2.5-7 [63] to analyze the statistics associated with the samples grouping. In detail, PERMANOVA (Adonis), ANOSIM and PERMDISP are all measures of beta-diversity and test different hypotheses. In order to provide complete information in terms of location and dispersion of our data we performed all the three tests. PERMANOVA tests if the centroids, similar to means, of each group are significantly different from each other. ANOSIM is a method that tests whether two or more groups of samples are significantly different (similar to Adonis). It provides a measure of similarity, and its statistic R is based on the difference of mean ranks between groups and within groups. Having both tests significant gives more strength to the hypothesis of different composition between groups. PERMDISP is a measure of dispersion (variances) of the groups. If significant, the two groups are not homogenously dispersed. PERMANOVA and PERMDISP can be used to rigorously identify location vs. dispersion effects, respectively, in the space of the chosen resemblance measure. In our case, PERMDISP is always not significant, except for the comparison CRC/HC, and also from the plot is clearly visible a different dispersion. ANOSIM is very sensitive to heterogeneity while PERMANOVA (Adonis) was found to be largely unaffected by heterogeneity in Anderson and Walsh’s simulations [64] for balanced designs (as in the case of our study, enrolling 20 CRC and 20 HC samples).

Differential abundance analysis was performed using DESeq2 package at each taxa level, previously removing unassigned phyla, Cyanobacteria/Chloroplast and *Propionibacterium acnes* species, identified as contaminant in negative controls (mean percentage of presence in the 6 negative controls: 35.8%), and keeping taxa having counts >1 in at least 60% of samples for the downstream analysis. The design of the model for each comparison included the Run variable. The *p*-values obtained by the DeSeq2 differential abundance analysis are corrected for multiple testing using the Benjamini and Hochberg method. Taxa were considered significantly differentially abundant if the associated adjusted *p*-value was <0.05. In order to define the associations between CCL2 blood concentration and some selected variables (i.e., CCL2 vs. Age, CCL2 vs. *F. nucleatum* abundance, CCL2 vs. *B. fragilis* abundance, CCL2 vs. *G. haemolysans* abundance, CCL2 vs. *S. intermedius* abundance) a linear multiple regression model was used, adding for each regression analysis the variable presence/absence of tumor. We considered the model validated if each time a variable was included in the model, its regression *p*-value was lower than 0.05 and the direction of effect consistent.

## 5. Conclusions

Our data, if confirmed in a larger than the present cohort, suggest the colon dysbiosis might contribute to CRC pathogenesis by inducing inflammation through the increase in bacterial species with known inflammatory potential. In particular, *Fusobacterium nucleatum*, also found more abundant in OW/OB_HC than in NW_HC microbiomes, might represent the putative link between obesity, inflammation and the previously reported increased CRC-risk in obese subjects. In conclusion, CRC-associated bacteria, together with the increased CCL2 circulating levels, could represent potential biomarkers to use in the CRC diagnosis and management.

## Figures and Tables

**Figure 1 ijms-22-10747-f001:**
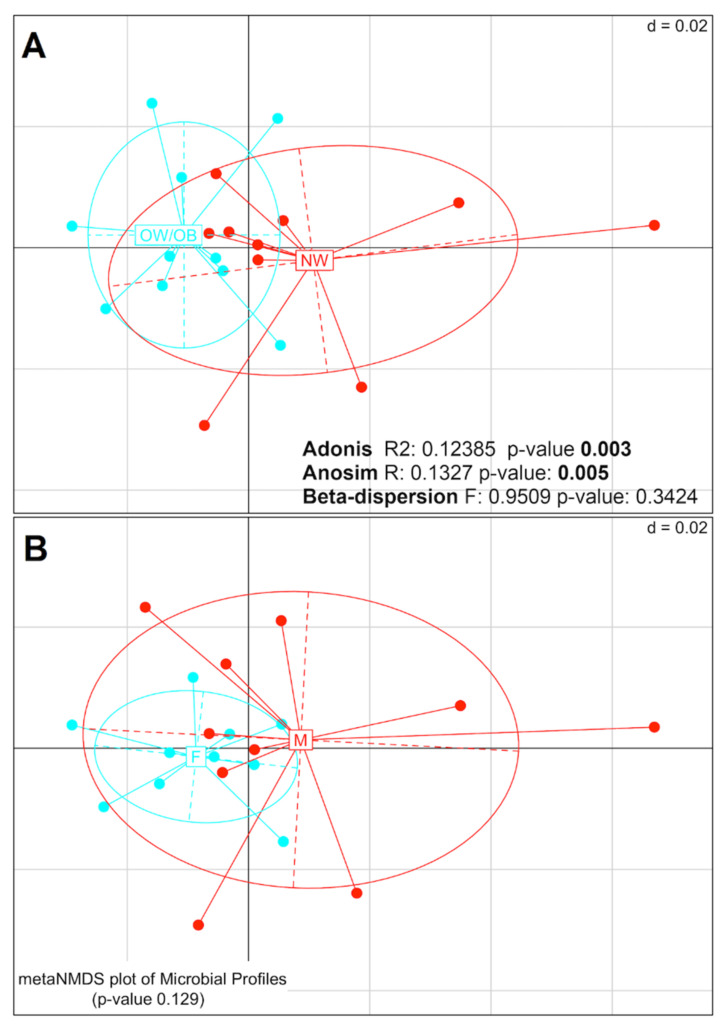
Nonmetric Multidimensional Scaling (NMDS) plots of weighted UniFrac distance to measure the beta diversity between (**A**) normal weight (NW) and overweight/obese (OW/OB), and (**B**) female and male healthy controls. The taxa counts table has been filtered keeping counts >1 in at least 10% of samples (383 taxa left) and then normalized through regularized logarithm transformation (DESeq2 package). Results of PERMANOVA (Adonis) test are shown in both panels. The significance in case of NW and OW/OB diversity (*p*-value = 0.003), indicates a separation between the centroids of the two groups. In addition, the values obtained through the Analysis of Similarity (ANOSIM) and beta-dispersion tests are shown. No significance is instead present grouping samples between females and males.

**Figure 2 ijms-22-10747-f002:**
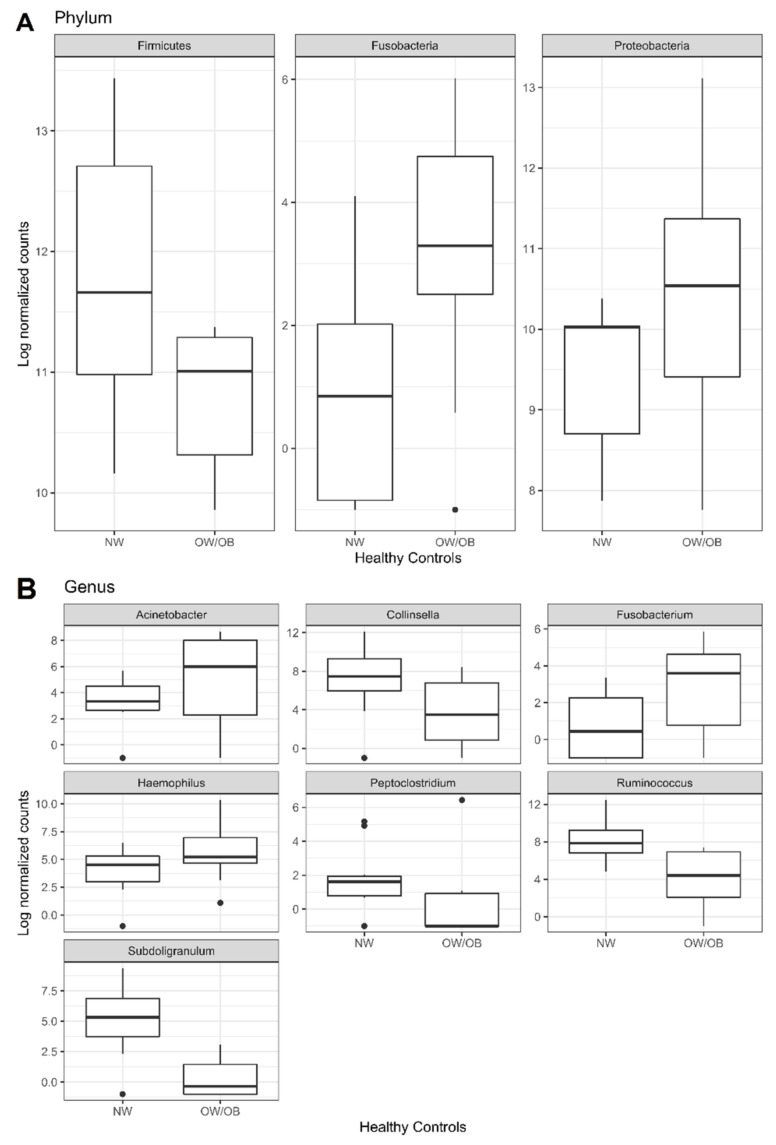
Boxplots of the significant results (adjusted *p*-value < 0.05) obtained through differential abundance test between OW/OB and NW healthy groups at (**A**) phylum and (**B**) genus levels. The test has been performed by applying DESeq2 Negative Binomial distribution. The taxa have been grouped by taxonomy level and those having counts >1 in at least 60% of samples have been kept. Lower and upper box boundaries 25th and 75th percentiles, respectively, line inside box median, lower and upper error lines 10th and 90th percentiles, respectively, filled circles data falling outside 10th and 90th percentiles.

**Figure 3 ijms-22-10747-f003:**
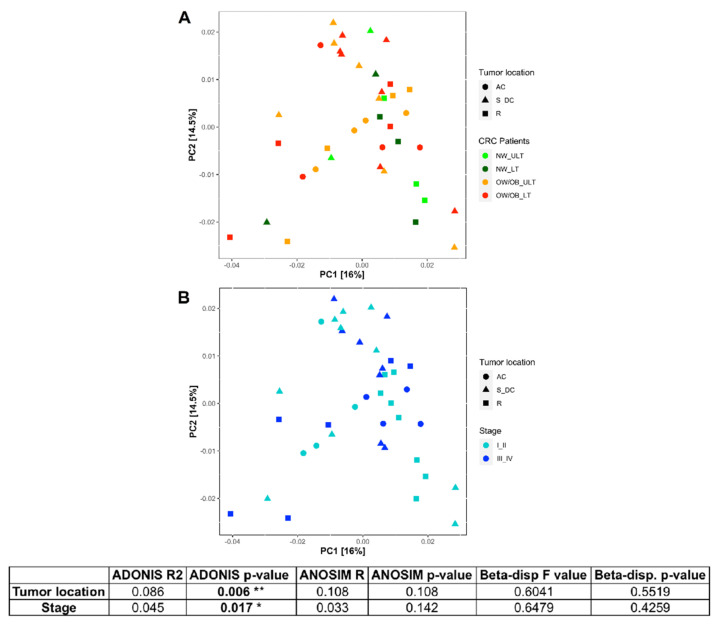
Principal Coordinate Analysis (PCoA) plots of weighted UniFrac distance showing the distribution of CRC patients: (**A**) colored by weight and lesion and shaped by tumor location and (**B**) colored by tumor stage and shaped by tumor location. The taxa counts table has been filtered keeping counts >1 in at least 10% of samples (508 taxa left) and then normalized through regularized logarithm transformation (DESeq2 package). The table shows the grouping factors for which PERMANOVA test (ADONIS) gave a *p*-value < 0.05 indicating a separation: tumor location and stage. Values obtained from Analysis of Similarity (ANOSIM) and beta-dispersion tests are also shown. AC, ascending colon; S_DC, sigma-descending colon; R, rectum. * *p* < 0.05; ** *p* < 0.001.

**Figure 4 ijms-22-10747-f004:**
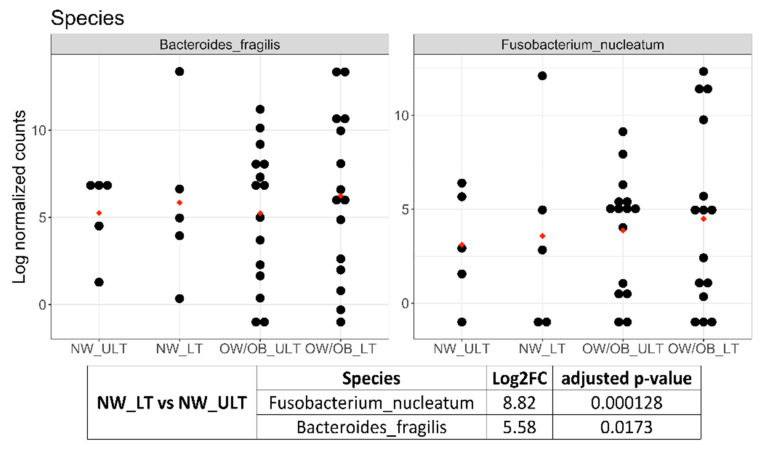
Dot plots of log transformed and DESeq2 normalized counts of *B. fragilis* and *F. nucleatum* in CRC samples grouped by weight and lesion. The red dots indicate the mean. The associated table shows the Log2FC and adjusted *p*-value of these two bacteria obtained by the comparison NW_LT vs. NW_ULT.

**Figure 5 ijms-22-10747-f005:**
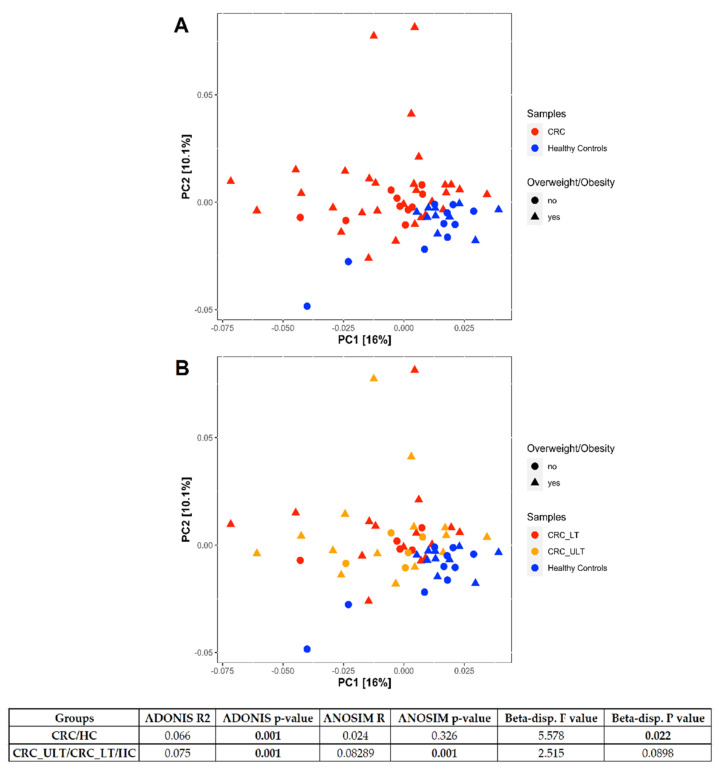
Principal Coordinate Analysis (PCoA) plots of weighted UniFrac distance showing the distribution of CRC patients and healthy controls. In (**A**), the samples are differently colored according to tumoral (CRC) or healthy (healthy controls) condition and the shapes indicate overweight/obesity comorbidity. In (**B**), the CRC samples are further differently colored according to tumor lesioned (LT) or unlesioned tissue (ULT). The taxa counts table has been filtered keeping counts >1 in at least 10% of samples (325 taxa left) and then normalized through regularized logarithm transformation (DESeq2 package). CRC patients and healthy controls are well separated, as also demonstrated by the significance obtained applying PERMANOVA (Adonis) test (*p* = 0.001). Values obtained from Analysis of Similarity (ANOSIM) and beta-dispersion tests are also shown.

**Figure 6 ijms-22-10747-f006:**
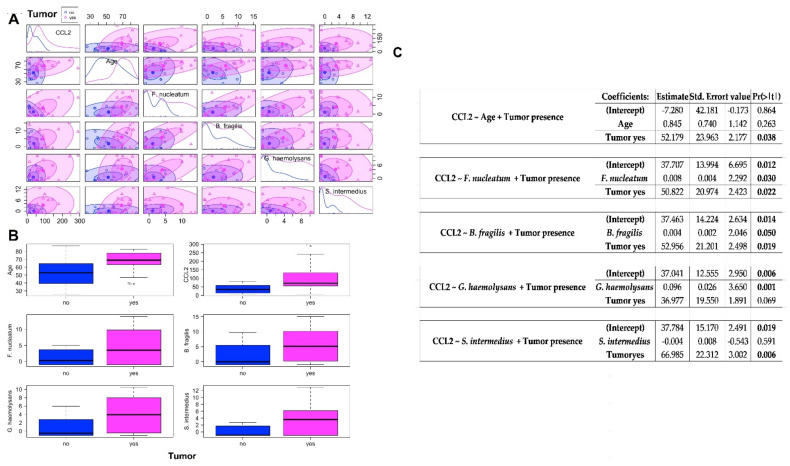
(**A**) Scatterplot matrix for circulating CCL2 levels (pg/mL), age (years), *F. nucleatum* abundance (log2 transformed and normalized), *B. fragilis* abundance (log2 transformed and normalized), *G. haemolysans* abundance (log2 transformed and normalized), S. intermedius abundance (log2 transformed and normalized) variables, showing within tumor presence groups (yes = CRC patients; no = healthy controls) 50 and 95% concentration ellipses. Inside the diagonal boxes the distribution curves are shown. (**B**) Boxplots for circulating CCL2 levels (pg/mL), age (years), *F. nucleatum* abundance (log2 transformed and normalized), *B. fragilis* abundance (log2 transformed and normalized), *G. haemolysans* abundance (log2 transformed and normalized), *S. intermedius* abundance (log2 transformed and normalized) variables in samples classified by tumor presence (yes = CRC patients; no = healthy controls). (**C**) Final model tables from multiple linear regression analysis in relation to the circulating CCL2 levels. Intercept = the mean for the response when all of the explanatory variables take on the value 0. Estimate = the estimated effect (also called the regression coefficient or r2 value), tells us that for every unit increase in microbial abundance there is an associated increase in CCL2 levels (the amount is given by the estimate corresponding to the bacterium), and that in case of CRC disease there is an associated increase in CCL2 levels. Std.error = standard error of the estimate. This number shows how much variation there is around the estimates of the regression coefficient. t value = statistic test. The statistic test used in linear regression is the t-value from a two-sided t-test. The larger the statistic test, the less likely it is that the results occurred by chance. Pr(>|t|) = *p*-value. This shows how likely the calculated t-value would have occurred by chance if the null hypothesis of no effect of the parameter were true.

**Table 1 ijms-22-10747-t001:** Differentially abundant taxa (6 taxonomic ranks) between OW/OB and NW healthy control samples with the associated Log2FC values and adjusted *p*-values.

Level	Taxa	Log2FC	Adjusted *p*-Value
Phylum	Firmicutes	−1.25	0.0072
	Fusobacteria	2.3	0.0282
	Proteobacteria	1.57	0.0134
Class	Clostridia (Firmicutes)	−2.15	0.0009
	Coriobacteriia (Actinobacteria)	−2.52	0.0250
	Fusobacteriia (Fusobacteria)	2.11	0.0444
	Gammaproteobacteria (Proteobacteria)	1.84	0.0143
Order	Bifidobacteriales (Actinobacteria)	−3.8	0.0002
	Clostridiales (Firmicutes)	−3.29	0.0001
	Coriobacteriales (Actinobacteria)	−4.06	0.0002
Family	Bifidobacteriaceae (Actinobacteria)	−3.53	0.0030
	Coriobacteriaceae (Actinobacteria)	−3.22	0.0051
	Ruminococcaceae (Firmicutes)	−3.22	0.0030
Genus	Acinetobacter (Proteobacteria)	3.14	0.0138
	Collinsella (Actinobacteria)	−3.3	0.0210
	Fusobacterium (Fusobacteria)	2.79	0.0418
	Haemophilus (Proteobacteria)	2.55	0.0210
	Peptoclostridium (Firmicutes)	−4.03	0.0138
	Ruminococcus (Firmicutes)	−4.2	0.0002
	Subdoligranulum (Firmicutes)	−6.31	0.0000
Species	*Acinetobacter lwoffii* (Proteobacteria)	4.17	0.0062
	*Bacteroides dorei* (Bacteroidetes)	−5.88	0.0002
	*Bifidobacterium longum* (Firmicutes)	−3.91	0.0051
	*Escherichia coli* (Proteobacteria)	2.62	0.0089
	*Fusobacterium nucleatum* (Fusobacteria)	3.26	0.0176
	*Haemophilus parainfluenzae* (Proteobacteria)	2.07	0.0487
	*Peptoclostridium difficile* (Firmicutes)	−3.26	0.0454
	*Ruminococcus torques* (Firmicutes)	−3.22	0.0089
	*Subdoligranulum variabile* (Firmicutes)	−4.67	0.0002

**Table 2 ijms-22-10747-t002:** Differentially abundant species in multiple comparisons performed analyzing CRC patients and healthy controls (HC) together, with the associated Log2FC values and adjusted *p*-values.

	Species	Log2FC	Adjusted *p*-Value
**CRC vs. Healthy Controls**	*Streptococcus intermedius* (Firmicutes)	5.72	9 × 10^−9^
	*Gemella haemolysans* (Firmicutes)	5.4	1.15 × 10^−8^
	*Ruminococcus torques* (Firmicutes)	−3	9.91 × 10^−5^
	*Fusobacterium nucleatum* (Fusobacteria)	3.88	5.65 × 10^−4^
	*Escherichia coli* (Proteobacteria)	1.85	5.65 × 10^−4^
	*Bacteroides fragilis* (Bacteroidetes)	2.44	1.43 × 10^−2^
	*Haemophilus parainfluenzae* (Proteobacteria)	2.11	4.60 × 10^−2^
**CRC_ULT vs. Healthy Controls**	*Gemella haemolysans*	4.84	3.76 × 10^−5^
	*Streptococcus intermedius*	4.72	5.52 × 10^−5^
	*Ruminococcus torques*	−3.21	3.24 × 10^−3^
	*Escherichia coli*	1.84	5.08 × 10^−3^
	*Fusobacterium nucleatum*	3.4	2.31 × 10^−2^
	*Ruminococcus gnavus* (Firmicutes)	2.03	4.85 × 10^−2^
**CRC_LT vs. Healthy Controls**	*Streptococcus intermedius*	7.06	2.65 × 10^−10^
	*Fusobacterium nucleatum*	7.33	5.33 × 10^−8^
	*Gemella haemolysans*	5.85	6.43 × 10^−8^
	*Bacteroides fragilis*	4.18	7.34 × 10^−4^
	*Haemophilus parainfluenzae*	−3.44	2.96 × 10^−3^
	*Escherichia coli*	1.85	3.23 × 10^−3^
	*Bacteroides vulgatus* (Bacteroidetes)	−2.03	4.52 × 10^−2^
**NW_ULT_CRC vs. NW_HC**	*Escherichia coli*	3.36	4.48 × 10^−3^
	*Ruminococcus torques*	−5.18	4.48 × 10^−3^
	*Gemella haemolysans*	4.63	4.74 × 10^−2^
**OW/OB_ULT_CRC vs. OW/OB_HC**	*Streptococcus intermedius*	7.72	1.78 × 10^−5^
	*Gemella haemolysans*	4.45	5.68 × 10^−3^

CRC = CRC_LT + CRC_ULT samples; CRC_LT = CRC patient tumor lesioned tissue; CRC_ULT = CRC patient unlesioned tissue, i.e., CRC adjacent normal tissue; HC = healthy controls.

## Data Availability

The dataset supporting the results of this article was deposited in the Sequence Read Archive (SRA) under BioProject accession code PRJNA736583.

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
