# Peer review of "16S rRNA of Mucosal Colon Microbiome and CCL2 Circulating Levels Are Potential Biomarkers in Colorectal Cancer"

_ijms, 2021, doi:10.3390/ijms221910747_

Round 1

Reviewer 1 Report

In their report, Nardelli et al investigate the differences in microbial content of colorectal cancer patients and normal controls, between "lesioned" and "unlesioned" tissue regions in colorectal patients, and between normal weight and overweight individuals. These results could be broadly informative and provide documentation of differences in the tumor microbiome and basis for discovery of early detection biomarkers. Limitations exist in english language, the large number of abbreviations, and questions about the statistical validity of some of the comparisons. 

  1. The manuscript should be edited by a native English speaker for clarity. There are numerous instances of odd phrasings and sentences that do not make sense. For example, the authors use the word “concur” or “concurring” frequently in inappropriate ways.
  2. Could the authors comment on the difference in average age (53 vs 69years) between the 2 study groups? This could confound some of the conclusions. Can the authors stratify groups by age to see if their findings could simply represent differences in microbiome based on age?
  3. Many terms are not defined in the results section (OTUs, Unifrac distance, Shannon diversity, Chao richness)
  4. It is unclear how the bacterial classifications are displayed in table 1 and throughout the manuscript. How do the genus/family/species information relate to the phylum/class information? For example, are all of the families listed a subset of one of the phyla?
  5. How were statistical analyses corrected for multiple hypothesis testing?
  6. What does “CRC_unlesioned” mean? Do these participants have colon cancer or not? It would be helpful to rename these groups, possibly to “CRC_tumor” and “CRC_adjacent_normal_tissue”. The word “lesion” is nonspecific and confusing.
  7. In Figure 3, there does not appear to be any separation of the samples based on stage or anatomic location in the PCA plots. This is in conflict with the authors claim that “the distributions of microbiomes in CRC patients were significantly separated on the basis of tumor location and stage”
  8. There are too many abbreviations to keep track of. It would be much clearer if the authors could write out the full names for anatomic locations. Some paragraphs are simply strings of abbreviations.
  9. In Figure 4, there does not appear to be differences between NW_LT and NW_ULT as described in the results section and table. There are only 5 samples and the distributions appear nearly identical between these groups. This raises concerns about the methods used to calculate statistical significance.
  10. What is the difference between CRC_ULT and “Normal controls”. The terminology used to designate these groups is confusing.
  11. Data in figure 5 would appear to be the most interesting and clinically informative information, but these results are not discussed in as much detail as the previous data.
  12. The table in Figure 6C lists a bunch of numbers but it is difficult to determine which of these are important and support the authors conclusions.
  13. Can the authors show individual graphs with data showing the primary data comparisons between CRC and healthy controls for several of the significant bacterial species? Currently only the summary data (fold change and p-value) are plotted.

Author Response

Point by point Reply to REVIEWER 1

Comments and Suggestions for Authors

In their report, Nardelli et al investigate the differences in microbial content of colorectal cancer patients and normal controls, between "lesioned" and "unlesioned" tissue regions in colorectal patients, and between normal weight and overweight individuals. These results could be broadly informative and provide documentation of differences in the tumor microbiome and basis for discovery of early detection biomarkers. Limitations exist in english language, the large number of abbreviations, and questions about the statistical validity of some of the comparisons. 

  1. The manuscript should be edited by a native English speaker for clarity. There are numerous instances of odd phrasings and sentences that do not make sense. For example, the authors use the word “concur” or “concurring” frequently in inappropriate ways.

We ameliorated the English, however, if the Editor thinks an English further revision is needed, we will be eager to pay the duty to the English revision office.

  1. Could the authors comment on the difference in average age (53 vs 69years) between the 2 study groups? This could confound some of the conclusions. Can the authors stratify groups by age to see if their findings could simply represent differences in microbiome based on age?

We are aware that age is one of the factors influencing the microbiome composition, but we choose to not stratify our groups by age being both groups limited in numerosity. Nevertheless, taking in account the Reviewer observation, we evaluated the trend of microbiome composition in our study groups aged < or ≥60 years at level of the main four phyla detected (see figure below only for Reviewer use). As it is shown, the taxa counts for each phylum were similar in both CRC and Healthy groups divided by age, even if we did not test the statistical significance considering the low number of subjects/group. Further, as we showed in Figure 6, CCL2 circulating levels were significantly associated with tumor presence (p=0.00343) and with the abundance of some bacteria regardless of the age of the study groups. Based on the above consideration we think the variations between CRC and Healthy subjects microbiomes observed in our study are not likely based on the different age of the enrolled subjects. However, we underlined the difference mean age of CRC patients and Healthy control subjects among the limitations of the study. See Discussion lines 571-572.

Figure only for the Reviewer use: Log normalized counts of bacterial taxa at phylum level in HC subjects and CRC patients aged < or ≥60 y (HC<60 y=13; CRC<60 y=4; HC≥60 y=7; CRC≥60 y=16).

  1. Many terms are not defined in the results section (OTUs, Unifrac distance, Shannon diversity, Chao richness)

We now better define in the Methods section the terms indicated from the Reviewer. See Methods pages 4, line 176; page 5 lines 195-198, 200-201, 203-207.

  1. It is unclear how the bacterial classifications are displayed in table 1 and throughout the manuscript. How do the genus/family/species information relate to the phylum/class information? For example, are all of the families listed a subset of one of the phyla?

We thank the Reviewer for his/her observation. We now added phylum information in Table 1, Table 2 and in the text page 9 lines 301-303.

  1. How were statistical analyses corrected for multiple hypothesis testing?

The p-values obtained by the DeSeq2 differential abundance analysis are corrected for multiple testing using the Benjamini and Hochberg method.

We added this information in Methods section. See page 5, lines 233-235.

  1. What does “CRC_unlesioned” mean? Do these participants have colon cancer or not? It would be helpful to rename these groups, possibly to “CRC_tumor” and “CRC_adjacent_normal_tissue”. The word “lesion” is nonspecific and confusing.

The “CRC unlesioned” indicates the non cancerous tissue nearby the cancer in CRC colon of patients. Considering this Reviewer’s observation and taking also in account the suggestion of another reviewer, we now better defined “CRC_LT” as CRC patient tumor lesioned tissue and “CRC_ULT” as CRC unlesioned tissue in the text.

See Introduction page 2 lines 87-88 and all the text.

  1. In Figure 3, there does not appear to be any separation of the samples based on stage or anatomic location in the PCA plots. This is in conflict with the authors claim that “the distributions of microbiomes in CRC patients were significantly separated on the basis of tumor location and stage”

Given the limitation of the bidimensional plots, where only two dimensions are shown, we tested the different location of data belonging to the two different groups performing PERMANOVA (Adonis). It tests if the centroids, similar to means, of each group are significantly different from each other. According the ADONIS test the distributions of microbiomes in CRC patients were significantly separated on the basis of tumor location and stage.

We add more details regarding the PERMANOVA (Adonis) test in the text page 10, lines 316, 317.

  1. There are too many abbreviations to keep track of. It would be much clearer if the authors could write out the full names for anatomic locations. Some paragraphs are simply strings of abbreviations.

We thank this Reviewer for the suggestion, we now added in the Figure 3 legend and in the text the full names of the abbreviations used for the anatomic locations. 

See page 10, line 311, 327; page 11 lines 333-335.

  1. In Figure 4, there does not appear to be differences between NW_LT and NW_ULT as described in the results section and table. There are only 5 samples and the distributions appear nearly identical between these groups. This raises concerns about the methods used to calculate statistical significance.

We agree with the Reviewer that although the means (red dots) are different, some samples show comparable values. We used the DeSeq2 package to determine the differential abundance and its significance. It exploits negative binomial generalized linear models; the estimates of dispersion and logarithmic fold changes incorporate data-driven prior distributions and it internally corrects the counts for sample size. It is a widely used approach, considered reliable and validated. Anyway, we are totally aware that the number of samples is limited and that our conclusions have to be taken as indications and need further experimental validation.

We now added the latter conclusion in discussion. See page 2 of the Discussion lines 476-478.

  1. What is the difference between CRC_ULT and “Normal controls”. The terminology used to designate these groups is confusing.

The abbreviation CRC_ULT indicates the microbiome found in CRC patients tumor unlesioned tissue, whereas Healthy control (HC) group indicates the microbiome found in the colon of subjects without cancer. NW_HC indicates Healthy controls with BMI<25 kg/m2, whereas OW/OB_HC indicates Healthy controls with BMI≥25 kg/m2.   

We now better detailed the terminology used in the text and in the Table 2. See page 13 Table 2, abstract and the text.

  1. Data in figure 5 would appear to be the most interesting and clinically informative information, but these results are not discussed in as much detail as the previous data.

We thank the Reviewer for his/her helpful suggestion and we now added in the Discussion further comments concerning the Figure 5.

See Discussion lines 505-507 and 510-513.

  1. The table in Figure 6C lists a bunch of numbers but it is difficult to determine which of these are important and support the authors conclusions.

We thank the Reviewer for his/her observations. We now better defined some details about the variables reported in Figure 6C and in the text.

“…..Intercept = the mean for the response when all of the explanatory variables take on the value 0. Estimate = the estimated effect (also called the regression coefficient or r2 value), tells us that for every unit increase in microbial abundance there's an associated increase in CCL2 levels (the amount is given by the estimate corresponding to the bacterium), and that in case of CRC disease there is an associated increase in CCL2 levels. Std.error = standard error of the estimate. This number shows how much variation there is around the estimates of the regression coefficient. t value = statistic test. The statistic test used in linear regression is the t-value from a two-sided t-test. The larger the statistic test, the less likely it is that the results occurred by chance. Pr( > | t | ) = p-value. This shows how likely the calculated t-value would have occurred by chance if the null hypothesis of no effect of the parameter were true”.

See Figure 6 legend page lines 410-415 and Results 3.2 section, page 14, lines 385-387, 393-400.

  1. Can the authors show individual graphs with data showing the primary data comparisons between CRC and healthy controls for several of the significant bacterial species? Currently only the summary data (fold change and p-value) are plotted.

We thank the Reviewer for his/her suggestion. We now show in the Supplementary Figure 4 the primary data comparisons between CRC and HC for the several significant bacterial species. Furthermore, we removed Bacteroides vulgatus from Table 2 being this latter not significantly differently abundant between CRC and HC and previously included by mistake.

See Supplementary Figure 4.

Reviewer 2 Report

Nardelli et al showed that mucosal colon microbiome and MCP1 circulating levels represent potential biomarkers in CRC and could contribute to the CRC management.. The paper is nicely presented and sound from a scientific point of view. Thus, I recommend it's publication

Author Response

We thank the Reviewer for his positive evaluation of our study.

Reviewer 3 Report

The study by Nardelli et al. is an excellent initial investigation into correlations between the colon microbiome and colon cancer and circulating CCL2 levels. However, the manuscript needs several changes to make it easier for the general reader to understand.

Suggested changes:
1. CCL2 should be used in place of MCP1: see, UniprotKB, GeneCards, and NCBI web pages.

2. The passage "... and 20 CRC patients, with/without overweight/obesity co-morbidity (OW/OB and NW, respectively)" is not clear. A more in-depth explanation should be used. For example "... and 20 CRC patients: 10 HC patients were classified as overweight/obese (OW/OB_HC) and 10 patients were normal weight (NW_HC); 15 CRC patients were classified as overweight/obese (OW/OB_CRC) and 5 patients were normal weight (NW_CRC)."

3. The passage "... were significantly increased in both lesioned and unlesioned CRC-microbiomes respect to HC-microbiome.". Again, A more in-depth explanation should be used. For example "... were significantly increased in CRC patient tumor/lesioned tissue (CRC_LT) and CRC patient unlesioned tissue (CRC_ULT) microbiomes with respect to HC microbiomes."

4. The abstract has two concluding statements. "Conclusions: Our data suggest the mucosal colon dysbiosis might concur to CRC pathogenesis by inducing inflammation. Further, Fusobacterium nucleatum, also found more abundant in OW/OB_HC than in NW_HC microbiome, might represent the putative link between obesity and the increased CRC-risk. In conclusion, mucosal colon microbiome and MCP1 circulating levels represent potential biomarkers in CRC and could contribute to the CRC management." The first statement more closely aligns with the data presented. 

5. Only the most important points and conclusions should be stated in the Abstract. Given the sample size, the age-independence of circulating CCL2 levels should not be stated in the Abstract. For example: "Background: Colorectal cancer (CRC) is one of the most common malignancies in the western world and intestinal dysbiosis might contribute to its pathogenesis. Methods: The mucosal colon microbiome and C-C motif chemokine 2 (CCL2), also known as Monocyte Chemoattractant Protein 1 (MCP1), were investigated in 20 healthy controls (HC) and 20 CRC patients using 16S rRNA-sequencing and immunoluminescent-assay, respectively. 10 HC patients were classified as overweight/obese (OW/OB_HC) and 10 patients were normal weight (NW_HC); 15 CRC patients were classified as overweight/obese (OW/OB_CRC) and 5 patients were normal weight (NW_CRC). Results: Fusobacterium nucleatum and Escherichia coli were more abundant in OW/OB_HC than in NW_HC microbiomes. Globally, Streptococcus intermedius, Gemella haemolysans, Fusobacterium nucleatum, Bacteroides fragilis and Escherichia coli were significantly increased in CRC patient tumor/lesioned tissue (CRC_LT) and CRC patient unlesioned tissue (CRC_ULT) microbiomes compared to HC microbiomes. MCP1 circulating levels were associated with tumor presence and with the abundance of Fusobacterium nucleatum, Bacteroides fragilis, and Gemella haemolysans. Conclusions: Our data suggest that mucosal colon dysbiosis might contribute to CRC pathogenesis by inducing inflammation. Notably, Fusobacterium nucleatum, which was more abundant in the OW/OB_HC than in the NW_HC microbiomes, might represent a putative link between obesity and increased CRC-risk."

In section 3.1, the concluding statement of the 1st paragraph should be "No statistically significant difference in the species richness, was detected in HC microbiomes when divided by BMI (NW < 25 kg/m2 or OW/OB ≥ 25 kg/m2) or by gender."
Also, in this sentence "species richness" should be defined. For example "species richness (number of species).
Therefore, this paragraph should read something like:
"In the present study we analyzed the microbial composition of the mucosal colon from healthy individuals (HC) and from CRC patients. First, we analyzed the microbial composition in HC. In Supplemental Figure 1 A, B the alpha diversity measured by Shannon diversity and Chao1 richness indices is shown. No statistically significant difference in species richness (number of species), was detected in normal weight HC (NW_HC) microbiomes compared to overweight/obese HC (OW/OB_HC) microbiomes: NW is defined as BMI < 25 kg/m2 and OW/OB is defined as BMI ≥ 25 kg/m2).  No statistically significant difference in species richness was detected in the microbiomes of females compared to males."

The second paragraph of section 3.1, is also very difficult for the general reader to follow. More explanation needs to be added to this paragraph and the conclusion statement should be the final statement of the paragraph. For example:
"The composition of the difference microbial groups (beta diversity analysis) was evaluated by PERMANOVA (Adonis), Anosim, and PERMDISP2 (Beta-dispersion) tests. Representation of sample distances through Nonmetric Multidimensional Scaling (NMDS) and the results of the associated statistical tests are shown in Figure 1A and B. The Adonis and Anosim tests indicate that the microbiomes of NW_HC and OW/OB_HC were significantly different (Adonis p-value=0.003; Anosim p-value=0.005), whereas no significant difference was  found using the Beta-dispersion test. No significant difference was found by any of the tests when grouping the samples by gender. Taken together, these analyses indicate that there is a significant difference in the compositions of difference microbial groups between NW_HC and OW/OB_HC, but not between females and males.
(Use Adonis rather than PERMANOVA here becasue Adonis is used in Fig. 1. Also, if a conclusion can be made from the different results of the different beta test, that conclusion should be stated here. Statements about weighted Unifrac distances showing OTU abundances should be in the Methods section.)

In the sentence "Differential abundance test highlighted the significantly different taxa (adjusted p-value < 0.05) between the OW/OB and NW HC samples from phylum to species level (Table 1)", I don't know what "adjusted" means. If p values were changes from the values given by the differential abundance analysis, the method of adjusting them and the rational for adjusting them should be given in the Methods. Also, the fact that Table 1 only presents taxa that were significantly different between the NW/HC and OW/OB_HC groups should be plainly stated. For example:
Differential abundance analysis highlighted the different taxa that were significantly different (adjusted p-value < 0.05) between the OW/OB_HC and NW_HC samples from phylum to species level (Table 1: note that taxa that were not significantly different between the OW/OB_HC and NW_HC samples are not listed).

Because ADONIS is used in Fig. 3, the phrase "(PERMANOVA test p<0.05) (Figure 3)" should be either "(PERMANOVA test p<0.05) (Figure 3)" or "(PERMANOVA [ADONIS] test p<0.05) (Figure 3)".

In Fig. 3, the ADONIS p-value = 0.017, the ANOSIM p-value = 0.142 and the Beta-disp p-value = 0.4259. Since the conclusion is that "the distributions of microbiomes in CRC patients were significantly separated on the basis of tumor location and stage (PERMANOVA test p<0.05) (Figure 3)", it appears that the ANOSIM and Beta-disp tests are not important. Why are they shown? Also, both the ANOSIM "R" and "p" values indicate that the ANOSIM test does not detect a difference between Tumor Location and Stage.

Because ADONIS is used in Fig. 5, the phrase "(PERMANOVA test p=0.001) (Figure 5A, B)" should be either "(PERMANOVA test p=0.001) (Figure 5A, B)" or "(PERMANOVA [ADONIS] test p=0.001) (Figure 5A, B)".

In Fig. 5, the p values obtained from the different tests are different. What is the meaning of these differences?

Section 3.2 states "Moreover, it is worth noticing that in the case of G. haemolysans the estimated increase is higher than the others bacteria, and, contrary to F. nucleatum and B. fragilis, for a fixed amount of G. haemolysans, changes regarding the presence of the tumor does not significantly affect MCP1 levels." I don' know what this statement means.

In the Discussion, "SCFA" needs to be defined: "short chain fatty acids (SCFA)".

Supplementary Table 1 should use "OW/OB_HC" as this is the order used to refer to overweight/obese healthy controls through out the manuscript.

The English grammar and diction needs to be corrected throughout the manuscript.

Author Response

Point by point Reply to REVIEWER 3

Comments and Suggestions for Authors

The study by Nardelli et al. is an excellent initial investigation into correlations between the colon microbiome and colon cancer and circulating CCL2 levels. However, the manuscript needs several changes to make it easier for the general reader to understand.

Suggested changes:
1. CCL2 should be used in place of MCP1: see, UniprotKB, GeneCards, and NCBI web pages.

We thank the Reviewer for his/her suggestion. We now reported CCL2 instead of MCP1 in the title, abstract and in the text.

  1. 2. The passage "... and 20 CRC patients, with/without overweight/obesity co-morbidity (OW/OB and NW, respectively)" is not clear. A more in-depth explanation should be used. For example "... and 20 CRC patients: 10 HC patients were classified as overweight/obese (OW/OB_HC) and 10 patients were normal weight (NW_HC); 15 CRC patients were classified as overweight/obese (OW/OB_CRC) and 5 patients were normal weight (NW_CRC)."
  2. 3. The passage "... were significantly increased in both lesioned and unlesioned CRC-microbiomes respect to HC-microbiome.". Again, A more in-depth explanation should be used. For example "... were significantly increased in CRC patient tumor/lesioned tissue (CRC_LT) and CRC patient unlesioned tissue (CRC_ULT) microbiomes with respect to HC microbiomes."
  3. 4. The abstract has two concluding statements. "Conclusions: Our data suggest the mucosal colon dysbiosis might concur to CRC pathogenesis by inducing inflammation. Further, Fusobacterium nucleatum, also found more abundant in OW/OB_HC than in NW_HC microbiome, might represent the putative link between obesity and the increased CRC-risk. In conclusion, mucosal colon microbiome and MCP1 circulating levels represent potential biomarkers in CRC and could contribute to the CRC management." The first statement more closely aligns with the data presented. 
  4. 5. Only the most important points and conclusions should be stated in the Abstract. Given the sample size, the age-independence of circulating CCL2 levels should not be stated in the Abstract. For example: "Background: Colorectal cancer (CRC) is one of the most common malignancies in the western world and intestinal dysbiosis might contribute to its pathogenesis. Methods: The mucosal colon microbiome and C-C motif chemokine 2 (CCL2), also known as Monocyte Chemoattractant Protein 1 (MCP1), were investigated in 20 healthy controls (HC) and 20 CRC patients using 16S rRNA-sequencing and immunoluminescent-assay, respectively. 10 HC patients were classified as overweight/obese (OW/OB_HC) and 10 patients were normal weight (NW_HC); 15 CRC patients were classified as overweight/obese (OW/OB_CRC) and 5 patients were normal weight (NW_CRC). Results: Fusobacterium nucleatum and Escherichia coli were more abundant in OW/OB_HC than in NW_HC microbiomes. Globally, Streptococcus intermedius, Gemella haemolysans, Fusobacterium nucleatum, Bacteroides fragilis and Escherichia coli were significantly increased in CRC patient tumor/lesioned tissue (CRC_LT) and CRC patient unlesioned tissue (CRC_ULT) microbiomes compared to HC microbiomes. MCP1 circulating levels were associated with tumor presence and with the abundance of Fusobacterium nucleatum, Bacteroides fragilis, and Gemella haemolysans. Conclusions: Our data suggest that mucosal colon dysbiosis might contribute to CRC pathogenesis by inducing inflammation. Notably, Fusobacterium nucleatum, which was more abundant in the OW/OB_HC than in the NW_HC microbiomes, might represent a putative link between obesity and increased CRC-risk."

Reply to points 2-5

We thank the Reviewer for his/her useful suggestions (points 2, 3, 4, 5) to make the abstract easier to understand and we modified the abstract accordingly.

“Background: Colorectal cancer (CRC) is one of the most common malignancies in the western world and intestinal dysbiosis might contribute to its pathogenesis. Methods: The mucosal colon microbiome and C-C motif chemokine 2 (CCL2) were investigated in 20 healthy controls (HC) and 20 CRC patients using 16S rRNA-sequencing and immunoluminescent-assay, respectively. Ten HC subjects were classified as overweight/obese (OW/OB_HC) and 10 subjects were normal weight (NW_HC); 15 CRC patients were classified as OW/OB_CRC and 5 patients were NW_CRC. Results: Fusobacterium nucleatum and Escherichia coli were more abundant in OW/OB_HC than in NW_HC microbiomes. Globally, Streptococcus intermedius, Gemella haemolysans, Fusobacterium nucleatum, Bacteroides fragilis and Escherichia coli were significantly increased in CRC patient tumor/lesioned tissue (CRC_LT) and CRC patient unlesioned tissue (CRC_ULT) microbiomes compared to HC microbiomes. CCL2 circulating levels were associated with tumor presence and with the abundance of Fusobacterium nucleatum, Bacteroides fragilis, and Gemella haemolysans. Conclusions: Our data suggest that mucosal colon dysbiosis might contribute to CRC pathogenesis by inducing inflammation. Notably, Fusobacterium nucleatum, which was more abundant in the OW/OB_HC than in the NW_HC microbiomes, might represent a putative link between obesity and increased CRC-risk”.

See Abstract page 1.

  1. In section 3.1, the concluding statement of the 1st paragraph should be "No statistically significant difference in the species richness, was detected in HC microbiomes when divided by BMI (NW < 25 kg/m2 or OW/OB ≥ 25 kg/m2) or by gender." Also, in this sentence "species richness" should be defined. For example "species richness (number of species). Therefore, this paragraph should read something like: "In the present study we analyzed the microbial composition of the mucosal colon from healthy individuals (HC) and from CRC patients. First, we analyzed the microbial composition in HC. In Supplemental Figure 1 A, B the alpha diversity measured by Shannon diversity and Chao1 richness indices is shown. No statistically significant difference in species richness (number of species), was detected in normal weight HC (NW_HC) microbiomes compared to overweight/obese HC (OW/OB_HC) microbiomes: NW is defined as BMI < 25 kg/m2 and OW/OB is defined as BMI ≥ 25 kg/m2).  No statistically significant difference in species richness was detected in the microbiomes of females compared to males."

We thank the Reviewer and modified the section 3.1 in Results as suggested. See Results page 6 lines 245-253.

  1. The second paragraph of section 3.1, is also very difficult for the general reader to follow. More explanation needs to be added to this paragraph and the conclusion statement should be the final statement of the paragraph. For example:
    "The composition of the difference microbial groups (beta diversity analysis) was evaluated by PERMANOVA (Adonis), Anosim, and PERMDISP2 (Beta-dispersion) tests. Representation of sample distances through Nonmetric Multidimensional Scaling (NMDS) and the results of the associated statistical tests are shown in Figure 1A and B. The Adonis and Anosim tests indicate that the microbiomes of NW_HC and OW/OB_HC were significantly different (Adonis p-value=0.003; Anosim p-value=0.005), whereas no significant difference was found using the Beta-dispersion test. No significant difference was found by any of the tests when grouping the samples by gender. Taken together, these analyses indicate that there is a significant difference in the compositions of difference microbial groups between NW_HC and OW/OB_HC, but not between females and males. (Use Adonis rather than PERMANOVA here becasue Adonis is used in Fig. 1. Also, if a conclusion can be made from the different results of the different beta test, that conclusion should be stated here. Statements about weighted Unifrac distances showing OTU abundances should be in the Methods section.).

We thank the Reviewer and modified the section 3.1 (second paragraph) in Results as suggested.

See Results page 6 lines 254-264.

We also added the statement about weighted Unifrac distances showing OTU abundances in Method section.

See methods page 5, lines 200-201, 203-207.

  1. In the sentence "Differential abundance test highlighted the significantly different taxa (adjusted p-value < 0.05) between the OW/OB and NW HC samples from phylum to species level (Table 1)", I don't know what "adjusted" means. If p values were changes from the values given by the differential abundance analysis, the method of adjusting them and the rational for adjusting them should be given in the Methods.

The p-values obtained by the DeSeq2 differential abundance analysis were corrected for multiple testing using the Benjamini and Hochberg method. We added this information in Methods section. See page 5, lines 233-235.

Also, the fact that Table 1 only presents taxa that were significantly different between the NW/HC and OW/OB_HC groups should be plainly stated. For example: Differential abundance analysis highlighted the different taxa that were significantly different (adjusted p-value < 0.05) between the OW/OB_HC and NW_HC samples from phylum to species level (Table 1: note that taxa that were not significantly different between the OW/OB_HC and NW_HC samples are not listed).

We modified the text according to the Reviewer suggestion. See page 7, lines 279-283.

  1. Because ADONIS is used in Fig. 3, the phrase "(PERMANOVA test p<0.05) (Figure 3)" should be either "(PERMANOVA test p<0.05) (Figure 3)" or "(PERMANOVA [ADONIS] test p<0.05) (Figure 3)".

We thank the Reviewer and now reported the PERMANOVA [ADONIS] test p<0.05 as suggested. See page 10, line 316.

  1. In Fig. 3, the ADONIS p-value = 0.017, the ANOSIM p-value = 0.142 and the Beta-disp p-value = 0.4259. Since the conclusion is that "the distributions of microbiomes in CRC patients were significantly separated on the basis of tumor location and stage (PERMANOVA test p<0.05) (Figure 3)", it appears that the ANOSIM and Beta-disp tests are not important. Why are they shown? Also, both the ANOSIM "R" and "p" values indicate that the ANOSIM test does not detect a difference between Tumor Location and Stage.

We thank the Reviewer and now better detailed in Methods (below) the different informations obtained by using the different statistical tests.

PERMANOVA (Adonis), ANOSIM and PERMDISP are all measures of beta-diversity and test different hypotheses. In order to provide complete information in terms of location and dispersion of our data we performed all the three tests. PERMANOVA tests if the centroids, similar to means, of each group are significantly different from each other. ANOSIM is a method that tests whether two or more groups of samples are significantly different (similar to adonis). It provides a measure of similarity and its statistic R is based on the difference of mean ranks between groups and within groups. Having both tests significant gives more strength to the hypothesis of different composition between groups. PERMDISP is a measure of dispersion (variances) of the groups. If significant, the two groups are not homogenously dispersed. PERMANOVA and PERMDISP can be used to rigorously identify location vs. dispersion effects, respectively, in the space of the chosen resemblance measure. In our case, PERMDISP is always not significant, except for the comparison CRC/HC, and also from the plot is clearly visible a different dispersion. ANOSIM is very sensitive to heterogeneity while PERMANOVA (Adonis) was found to be largely unaffected by heterogeneity in Anderson & Walsh's simulations (https://doi.org/10.1890/12-2010.1) for balanced designs (as in the case of our study enrolling 20 CRC and 20 HC samples)”.

See Method section page 5, lines 211-227.

  1. Because ADONIS is used in Fig. 5, the phrase "(PERMANOVA test p=0.001) (Figure 5A, B)" should be either "(PERMANOVA test p=0.001) (Figure 5A, B)" or "(PERMANOVA [ADONIS] test p=0.001) (Figure 5A, B)".

We thank the Reviewer and now reported the PERMANOVA [ADONIS] test p<0.05 as suggested. See page 11, line 349.

  1. In Fig. 5, the p values obtained from the different tests are different. What is the meaning of these differences?

We thank the Reviewer for the observation. We now added the meaning of the results obtained by the different statistical tests in Results section. See page 11, lines 349-352.

  1. Section 3.2 states "Moreover, it is worth noticing that in the case of G. haemolysans the estimated increase is higher than the others bacteria, and, contrary to F. nucleatum and B. fragilis, for a fixed amount of G. haemolysans, changes regarding the presence of the tumor does not significantly affect MCP1 levels." I don' know what this statement means.

We changed the sentence making it easier to understand than the previous one. See page 14, lines 395-400.

In the Discussion, "SCFA" needs to be defined: "short chain fatty acids (SCFA)".

We added the definition of the SCFA in the text. See line 459.

Supplementary Table 1 should use "OW/OB_HC" as this is the order used to refer to overweight/obese healthy controls through out the manuscript.

We thank again the Reviewer. We modified as suggested the Supplementary Table 1.

The English grammar and diction needs to be corrected throughout the manuscript.
We ameliorated the English, however, if the Editor thinks an English further revision is needed, we will be eager to pay the duty to the English revision office.

This manuscript is a resubmission of an earlier submission. The following is a list of the peer review reports and author responses from that submission.

Round 1

Reviewer 1 Report

In this manuscript, Nardelli C. et al., study the different microbial abundance in CRC and healthy individuals and correlate this finding with levels MCP-1 chemokine. It is an informative study and I support this manuscript to be published in Cancers journal. I have few minor comments that could improve the presentation of the study.

1) Why the authors chose specifically to check the levels of MCP-1 chemokine. Did they also check other cytokine and chemokine levels. Maybe they could refer to the literature or provide the data (if possible) that lead to the conclusion that among other cytokines/chemokines MCP-1 is the most relevant to be used as a diagnostic marker.

2) According to existing literature, MCP-1 is correlated with CRC metastasis (Role of MCP-1 in Alcohol-induced Aggressiveness of Colorectal Cancer Cells, Xu M. et al., 2016). It would be interesting that the authors provide more data on the metastatic events of the subjects under study (if any) and indicate if there is correlation with the MCP-1 levels.

Reviewer 2 Report

     Nardelli et al. have examined the gut microbiome profiles in CRC patients (n = 20) and healthy controls (n = 20). Subsequently, they isolate some candidate strains whose abundance is significantly different between these two groups. On the other hand, they also investigated the level of blood MCP1 in healthy controls and CRC patients and showed that MCP1 level is significantly increased in CRC patients. The following points should be addressed.

1. Throughout the manuscript, the letters in Figures are too small. Re-make the letters as possible as big.

2. Authors sub-analyzed their data by dividing healthy controls or CRC patients in terms of tumor location, obese, tumor or lesion/non-tumor lesion. Researches are interested in these sub-analyses. However, the number in each group becomes fewer. Thus, the obtained data is too primitive. Authors must know this point and describe the limitations.

3. Authors investigated the level of blood MCP1 in healthy controls and CRC patients. They used this molecule as a marker for inflammation. There are numerous markers for inflammation. Why did authors select MCP1? Moreover, blood MCP1 never reflect the inflammatory condition in the colonic tissues/tumor.

4. Serum MCP1 or plasma MCP1?

5. Among the bacterial strains isolated, Fusobacterium nucleatum is attracting much attention. Regarding Fusobacterium nucleatum, are there any interesting information, for example, location or histological type (SSA/P etc.).

Reviewer 3 Report

Nardelli et al.,’s manuscripts “Mucosal colon microbiome and MCP1 circulating levels are  potential biomarkers in colorectal cancer” investigated gut dysbiosis in healthy controls (HC) and  CRC patients, with/without overweight/obesity by 16S rRNA-sequencing. Also,  the monocyte chemoattractant protein 1 (MCP1) inflammatory cytokine levels were investigated associated in the two groups.

This study claimed that five bacteria: Streptococcus intermedius, Gemella haemolysans, Fusobacterium  nucleatum, Bacteroides fragilis and Escherichia coli, were significantly higher in the CRC-associated mucosal colon microbiome.  Moreover, the authors concluded that MCP1 circulating levels were,  independently from the age, significantly associated with the tumor presence and with the abundances of Fusobacterium nucleatum, Bacteroides fragilis and Gemella haemolysans. After that, the authors concluded that mucosal colon microbiome and MCP1 circulating levels represent potential biomarkers in CRC and could contribute to CRC management.

Accordingly, although this study is straightforward, the analysis is insufficient, and the results are not solid due to the improper analysis approaches used. The result can not be supported by current data. On the other hand, the figures were not generated well by the authors, and the discussion was not inadequate. Therefore, the idea is attractive, but unfortunately,  the authors draw firm conclusions from not convincing data.

Major points:

  1. The logFC analysis is not strong enough to show the differentially abundant species between groups. LEfSe (Linear discriminant analysis effect size) analysis is required to find biomarkers or stains between control and CRC groups.
  2. To link the biomarkers, the gene function/gene ontology and KEGG pathway of those enriched/differentially abundant species should also be analyzed.
  3. Frankly speaking, I can not see any significant separation from the Principal Coordinate Analysis (Fig.3 and5).
  4. The resolution of Fig.2 is low, hard to see the critical information.
  5. Another weakness of this study is the association analysis between MCP1 blood levels in CRC patients and the abundance of bacteria. I would like to strongly recommend the author to do the Spearman or Pearson correlation coefficient analysis. After that,  the conclusion may be more robust than now. I would like to see similar results as Figure7 shown in the references (https://doi.org/10.3389/fmicb.2018.01988). 
  6. The metabolism analysis of blood samples will enhance the manuscript.

Reviewer 4 Report

The authors have investigated Mucosal colon microbiome and MCP1 circulating levels are potential biomarkers in colorectal cancer and found mucosal colon microbiome and MCP1 circulating levels represent potential biomarkers in CRC and could contribute to CRC management. This topic is very interesting, the results were very significantly changed, and the study is very meaningful for the healthcare of patients. However, the patient sample sizes are relatively small and the average age of the two groups has a big difference – patients in the HC group are 16.2 years younger than those in CRC patients! It’s very known that the digestion system in the seniors has a big change so do the bacteria in the gut. I am not quite sure how much this contributes to the effects on the results. In addition, only one indicator, MCP1, made the results less fundamental.